# Control of Eukaryotic DNA Replication Initiation—Mechanisms to Ensure Smooth Transitions

**DOI:** 10.3390/genes10020099

**Published:** 2019-01-29

**Authors:** Karl-Uwe Reusswig, Boris Pfander

**Affiliations:** Max Planck Institute of Biochemistry, DNA Replication and Genome Integrity, 82152 Martinsried, Germany; kureusswig@biochem.mpg.de

**Keywords:** DNA replication, DNA replication initiation, cell cycle, post-translational protein modification, protein degradation, cell cycle transitions

## Abstract

DNA replication differs from most other processes in biology in that any error will irreversibly change the nature of the cellular progeny. DNA replication initiation, therefore, is exquisitely controlled. Deregulation of this control can result in over-replication characterized by repeated initiation events at the same replication origin. Over-replication induces DNA damage and causes genomic instability. The principal mechanism counteracting over-replication in eukaryotes is a division of replication initiation into two steps—licensing and firing—which are temporally separated and occur at distinct cell cycle phases. Here, we review this temporal replication control with a specific focus on mechanisms ensuring the faultless transition between licensing and firing phases.

## 1. Introduction

DNA replication control occurs with exceptional accuracy to keep genetic information stable over as many as 10^16^ cell divisions (estimations based on [1]) during, for example, an average human lifespan. A fundamental part of the DNA replication control system is dedicated to ensure that the genome is replicated exactly once per cell cycle. If this control falters, deregulated replication initiation occurs, which leads to parts of the genome becoming replicated more than once per cell cycle (reviewed in [2,3,4]). This over-replication (also referred to as rereplication) is accompanied by DNA damage, for example, in the form of double-strand breaks and subsequent genome rearrangements [5,6,7,8]. Notably, deregulation of DNA replication is increasingly recognized as a critical factor for genome instability during cancer development [9,10,11,12,13] and many oncogenes act by abrogating the G1/S cell cycle transition, thereby influencing DNA replication control [14,15]. Therefore, it seems plausible that over-replication could be a source of the genome rearrangements observed in tumour cells [16], but whether over-replication is a major driver of carcinogenesis is a question for future research.

Previous research has given us a detailed picture of the principal regulation of DNA replication initiation (see below) in the yeast *Saccharomyces cerevisiae* and also in metazoan systems. Replication initiation occurs in two interconnected steps—licensing and firing—which are coupled to separate phases of the cell cycle. Experimental systems to entirely abolish this separation cause widespread over-replication, a highly toxic condition. It is still a matter of active research as to how mutual exclusivity of licensing and firing is maintained at cell cycle transitions and, thus, how cells are protected from sporadic over-replication at these transitions. With this review, we aim to highlight established and also putative mechanisms that might act to ensure robust separation of licensing and firing and thus robustly block over-replication. We refer readers to the following excellent reviews for a detailed overview of the mechanism of replication initiation [2,17,18], elongation [18,19], and termination [18,20,21], as well as replication fork stalling [22,23,24]. 

## 2. DNA Replication Initiation in Eukaryotes

In eukaryotes, DNA replication initiates at many sites within the genome (replication origins) in parallel to allow fast duplication of large genomes. This brings about a need for tight control of initiation in order to ensure that each part of the genome is replicated exactly once per cell cycle. Cells achieve “once-per-cell-cycle replication initiation” by dividing the replication initiation process into two temporally separate phases—licensing and firing [2,3]. In mechanistic terms, licensing corresponds to the loading of inactive precursors of the Mcm2–7 helicase at replication origins by the pre-replicative complex ([25,26,27,28,29], Figure 1A, upper panel), while firing corresponds to activation of the replicative helicase by association of additional accessory subunits ([30,31,32,33,34,35,36], Figure 1A, lower panel). Previous studies have revealed the essential licensing and firing factors of budding yeast, and an in vitro reconstitution of origin-dependent initiation of replication has been achieved using the corresponding set of purified proteins [30,37,38,39,40]. In brief, licensing involves the licensing factors ORC (origin recognition complex Orc1–6), Cdc6, and Mcm2–7/Cdt1 and achieves origin recognition and ATP-dependent loading of the Mcm2–7 helicase core in the form of an inactive double hexamer, which encircles double-stranded DNA and is positioned in a head-to-head orientation, thus establishing bidirectionality of DNA replication (Figure 1A, [25,26,27,28,29,41,42,43,44,45,46,47]). Firing involves the helicase accessory subunits Cdc45 and GINS; the firing factors Sld2, Sld3, and Dpb11, as well as DNA polymerase ε and Mcm10 and achieves association of Cdc45 and GINS with Mcm2–7 and, thereby, activation of the replicative CMG helicase (Cdc45 Mcm2–7 GINS), remodeling of the helicase to encircle single-stranded DNA (the leading strand template), and initial DNA unwinding [36,37,48,49,50,51,52,53,54,55,56]. After this committed step of initiation, multiple replication factors such as DNA polymerases associate with the replicative CMG helicase to catalyze chromosome replication [18,19]. Notably, firing and licensing factors are conserved from yeast to human [57], suggesting that not only the principal mechanism of replication initiation is highly conserved during evolution, but also that these conserved factors will most likely be essential targets of control.

### 2.1. DNA Replication Initiation Control in Budding Yeast

Eukaryotic DNA replication initiates at multiple origins spread across the genome in order to allow a fast S phase despite large genomes. Features that define replication origins differ between species and have been comprehensively reviewed elsewhere [58]. Usage of multiple initiation sites inevitably brings with it the need for coordination. In particular, eukaryotic DNA replication control serves the purpose of generating a complete copy of the genome while avoiding any form of over-replication. Therefore, the two steps of initiation are interconnected (firing requiring prior licensing) but coupled to separate cell cycle phases, ensuring that every origin initiates at maximum once per cell cycle. Moreover, Mcm2–7 helicase precursors (the product of the licensing reaction) are removed from an origin when this origin is passively replicated [25,59,60], ensuring that origin firing cannot occur on post-replicative chromatin.

Temporal separation of licensing and firing, therefore, is key for ensuring that DNA replication at a given origin occurs only once per cell cycle. Indeed, when licensing and firing are experimentally induced to occur simultaneously, successive rounds of licensing and firing reactions trigger over-replication [5]. Temporal separation of licensing and firing is achieved by coupling them to specific phases of the cell cycle. Licensing generally occurs from late M phase to the G1/S transition [28,61,62]. Firing occurs in S phase, but the cellular firing potential (at least of budding yeast cells) remains high even until the metaphase-to-anaphase transition [31]. The next paragraphs focus on the well-understood cell cycle regulation of replication initiation in budding yeast, which is facilitated by the two major cell cycle kinases cyclin-dependent kinase 1 (Cdk1, also Cdc28, in the following referred to as CDK) and Dbf4-dependent kinase (DDK). Both kinases are regulated by the controlled expression and, importantly, destruction of regulatory subunits (cyclins and Dbf4, respectively) and are active from the G1/S transition to the metaphase-to-anaphase transition [63,64,65,66,67,68,69,70,71]. 

Firing essentially requires the activity of both CDK and DDK [72,73,74] and, indeed, many replication factors have been shown to be substrates of CDK and/or DDK phosphorylation [75,76,77,78,79,80]. Most importantly, the essential CDK and DDK substrates in replication initiation of budding yeast have been identified and specific mutants have been generated which bypass the CDK- and DDK-dependent steps, thus giving critical mechanistic insights into firing activation [48,49,51,81,82,83]. CDK phosphorylates Sld2 and Sld3, and this phosphorylation generates binding sites for Dpb11 [48,49,51,84,85]. Notably, Dpb11 contains two separate phosphoprotein binding sites [86,87,88,89,90,91,92,93], enabling it to simultaneously interact with Sld2 and Sld3 [48,49,84]. Although we currently lack insights into the precise architecture of the assembly of firing factors with the Mcm2–7 helicase, it is clear that the CDK-mediated interactions of Sld2, Sld3, and Dpb11 are critical to facilitate the association of the helicase accessory factors Cdc45 and GINS with Mcm2–7 and, thereby, the formation of the CMG helicase [84,94,95]. In particular, Cdc45 was suggested to be recruited in the form of a Sld3–Cdc45 complex [94,95,96,97], while GINS was suggested to be reeled in via Sld2 [84,98]. DDK, on the other hand, targets Mcm2–7 itself. DDK phosphorylation appears to not only relieve the Mcm2–7 complex from autoinhibition [81,82,83], it also appears to generate a Sld3-binding site on the Mcm2–7 complex [97]. Overall, the dependency on CDK and DDK therefore strictly links firing to S, G2, and early M phases [31].

Licensing, on the other hand, is inhibited by CDK phosphorylation of licensing factors. CDK phosphorylates ORC (Orc2 and Orc6 in particular), and this phosphorylation leads to inhibition of ORC [5,99]. CDK phosphorylates Cdc6 and the phosphorylation marks generate a degron on Cdc6, which facilitates destruction of Cdc6 by the ubiquitin–proteasome system [100,101,102,103]. CDK also phosphorylates the Mcm2–7 complex (in particular, Mcm3), which leads to the nuclear export of the soluble Mcm2–7/Cdt1 complex [104], with nuclear degradation of Mcm3 seemingly playing a backup role [105]. 

Furthermore, CDK also inhibits licensing in phosphorylation-independent ways. In particular, CDK was shown to engage in inhibitory protein–protein interactions with both Cdc6 (Clb2–Cdk1) and ORC [106,107,108]. The inhibition of the licensing reaction by CDK therefore restricts licensing to the cellular state of CDK inactivity from late mitosis to the G1/S transition. 

### 2.2. Additional DNA Replication Initiation Control Mechanisms in Metazoa

The principles of replication control are highly similar in different eukaryotes and, at this point, it appears certain that division of replication initiation into separate licensing and firing phases occurring in G1 and S is a universal feature of eukaryotes. However, the specific mechanisms restricting licensing and firing to different phases of the cell cycle appear to be much less conserved compared to the core mechanisms of DNA replication itself [57,109]. As such, it is not surprising that metazoa—while apparently still using the CDK- and DDK-dependent control of licensing and firing [110,111,112,113,114]—have developed additional mechanisms which are independent of phosphorylation events. Interestingly, at least two of these mechanisms impinge on the licensing factor Cdt1, making it a focal point of regulation.

The first mechanism involves the licensing inhibitor geminin, orthologs of which can be found in worms, insects, and vertebrates [115,116,117,118,119]. Geminin from *Xenopus laevis* binds to the licensing factor Cdt1, thereby sequestering it from its licensing role to promote Mcm2–7 loading [115,116]. Geminin is, in principle, CDK-independent but nonetheless under strict cell cycle control, giving cell cycle phase specificity to this mechanism of licensing inhibition. Geminin is an ubiquitylation substrate of APC^Cdh1^ and is degraded in late M and G1 [117]. Conversely, the presence of geminin leads to sequestration of Cdt1 from S to early M and to geminin-dependent suppression of licensing during this time. Therefore, this mechanism contributes to restricting licensing to late M and G1 as well as ensuring mutual exclusivity of licensing and firing.

A second example of licensing inhibition is Cdt1 degradation by the Cul4–Cdt2 ubiquitin ligase complex [120,121,122,123,124]. This mechanism is found in metazoa [120,121] as well as *Schizosaccharomyces pombe* [125,126] and is indirectly coupled to the cell cycle, as it is dependent on ongoing DNA replication. Specifically, Cul4–Cdt2 requires association with the replication elongation factor PCNA [122,123,124], which thereby functions as a replication-coupled platform for Cdt1 destruction and, thereby, licensing inhibition. Per definition, this mechanism will only be active when DNA replication elongation is ongoing. As such, it is suited to keep Cdt1 levels low during S phase, thereby contributing to licensing and firing separation, but it is insufficient to establish separation of licensing and firing on its own.

### 2.3. Deregulation of DNA Replication Initiation—Over-Replication and Genome Instability

The principles of replication control are highly similar in different eukaryotes, and temporal separation of replication initiation into two phases appears to be a universal feature of eukaryotic replication. In order to understand the consequences of deregulated replication initiation, several experimental systems have been developed. In particular, budding yeast cells as well as *Xenopus laevis* egg extracts have been insightful models and have collectively shown that a temporal overlap of licensing and firing leads to over-replication (see below). Over-replication, in turn, will cause DNA damage in the form of double-strand breaks (DSBs) and induce genome instability [7,8,127,128].

In budding yeast, the best-understood system to induce widespread over-replication utilizes mutants that bypass CDK-dependent controls of licensing, which otherwise block this process in S, G2, and early M. Specifically, CDK-dependent inhibition of licensing is abolished by (i) mutation of the CDK phosphorylation sites on Orc2 and Orc6, (ii) overexpression of a degradation-resistant, truncated version of Cdc6, and (iii) expression of Mcm7 with a constitutive nuclear localization signal [5]. Continued expression of these inhibition-resistant licensing factors is lethal for cells, underlining the importance of temporal separation of licensing and firing, but conditional systems have allowed studying over-replication by inducing licensing and over-replication in M or S phase cells, respectively [5,129]. Specific induction of over-replication in a single cell cycle using this system or derivatives, where only a subset of factors has been deregulated, has allowed initial insights into the consequences of over-replication [6,129,130,131,132]. From this work, we can conclude that over-replication is a potent inducer of genome instability, which manifests as chromosome rearrangements, gene amplifications, chromosomal instability, and even aneuploidy [6,8,130,132,133]. A major source of this genome instability appear to be DSBs in the over-replicated region which, given the inherent overamplification of genomic material, can only be repaired in a manner that involves rearrangements and/or duplications of chromosomal sequences. Currently, we cannot, however, say by which specific mechanism these DSBs are induced and whether other DNA lesions or structures contribute significantly to genome instability after over-replication, and we refer readers to excellent reviews dedicated to this topic [134,135].

Deregulation of licensing has also been the key to study the consequences of widespread over-replication in metazoan systems, in particular using the cell-free *Xenopus laevis* egg extract system. Strategies have involved depletion of geminin and inhibition of Cdt1 proteolysis [120,121,136,137] and exogenous addition of Cdt1 at high levels [136,138] to replicating extracts, as well as depletion of geminin [139,140,141] or the mitotic regulator Emi1 [128,142] in cultured cells. All systems lead to over-replication and also to the induction of DSBs, which trigger a cell cycle arrest that is dependent on the DNA damage checkpoint [128,141,143]. Collectively, these studies have also put forward the idea that the frequency of over-replication initiation events may be an important factor influencing the number of induced DSBs, but perhaps also the mechanism by which they are induced. 

### 2.4. Partial Deregulation of DNA Replication Initiation—Sporadic Over-Replication

While systems to induce widespread over-replication have been immensely informative, it can be argued for at least two reasons that sporadic over-replication is perhaps an even more relevant scenario in the context of cellular physiology: (i) Sporadic over-replication will not be as toxic as widespread over-replication. Not only will fewer over-replication events generate less DNA damage, less DNA damage will also not be as efficient in inducing the DNA damage checkpoint and its associated protective functions. Therefore, sporadic over-replication might manifest primarily in the form of genome instability and could be a contributing factor during tumorigenesis, when cells face deregulation of critical G1/S cell cycle controls [9,11,14,15]. (ii) A remarkable feature of DNA replication control is its multi-layered character; for example, CDK independently inhibits several licensing factors in budding yeast [5]. It is conceivable that such synergizing pathways confer overall robustness and, indeed, it has been suggested that overlapping pathways are required to give each replication origin the immense accuracy of replication initiation, which is statistically required for genome replication to be overall accurate [144]. Disabling just one layer of regulation will therefore generally only result in sporadic over-replication, but mutations disabling just one layer of control will be much more frequent compared to mutations disabling the entire control system.

While experimental systems to induce sporadic over-replication by partial deregulation of licensing [6] or firing factors [96,145] have been explored, our understanding of sporadic over-replication is limited by the lack of sensitive methods to analyze such events. While genetic and sequencing-based analyses of genome rearrangements are extremely sensitive, genome rearrangements can have many sources and are not uniquely linked to over-replication [146]. In contrast, commonly used physical assays are unable to visualize sporadic over-replication [5,129,147], and development of more sensitive methods will be a key future task for the field. Most importantly, our inability to specifically measure sporadic over-replication also means that we are currently unable to predict whether the principal set of replication control mechanisms has been identified or not. In the following section, we explore what kind of additional mechanisms ensuring a temporal separation of licensing and firing exist on top of the principal CDK/DDK control of replication initiation. In particular, we focus on the transitions between licensing and firing phases and on mechanisms that act at these cell cycle transitions in order to ensure robust separation of licensing and firing and, thereby, once-per-cell-cycle replication.

## 3. DNA Replication Control at Cell Cycle Transitions 

In order to prevent sporadic over-replication, cells need to avoid any scenario in which even sub-pools of licensing and firing factors are simultaneously active. As such, it is interesting to consider how such a scenario is avoided at cell cycle transitions and whether the principal cell cycle control mechanisms at these transitions are sufficient to guarantee mutual exclusivity or whether additional replication-specific mechanisms are employed. Importantly, the transitions between the two replication phases coincide with the two major cell cycle transitions, where cells switch between CDK-on and CDK-off states. 

### 3.1. Bistable Switches—The Fundament of DNA Replication Control at Cell Cycle Transitions

The problem of accurately separating events of one cell cycle phase from those in the next cell cycle phase is by no means a problem unique to DNA replication control. Indeed, sharp (ultrasensitive) and unidirectional cell cycle transitions are the very fundament of the cell cycle [148]. In a simplified model of the cell cycle, cells exist in one of two stable states—CDK-on and CDK-off—and the transitions between these two states are extremely rapid (switch-like). Progress in quantitative measurements of cell cycle factors combined with mathematical modeling shows that both the G1/S transition, during which cells switch to the CDK-on state and activate firing, as well as the metaphase-to-anaphase transition, during which cells switch to the CDK-off state and activate licensing, take the form of bistable switches [149,150,151]. These cell cycle transitions are also characterized by hysteresis and, thereby, are unidirectional and irreversible. On a molecular level, critical mechanisms that cause bistability and hysteresis are (i) positive feedback (specifically, double-negative feedback loops involving inhibitors of CDK (G1/S) or the APC (metaphase to anaphase) [152,153,154,155,156], (ii) complex, irreversible molecular processes such as protein degradation or translation, and (iii) post-translational modifications of key proteins occurring at multiple sites [157].

It is therefore logical to assume that the sharp and irreversible rise (G1/S) and drop (metaphase to anaphase) of CDK activity is the principal mechanism that causes rapid activation/inactivation of licensing and firing factors and thereby counteracts over-replication at cell cycle transitions. Indeed, a shallower rise of CDK activity at the G1/S transition (in budding yeast *sic1–cpd* mutants) coincides with enhanced chromosome loss [157], which can be interpreted to be a consequence of sporadic over-replication at the G1/S transition. Notwithstanding these initial insights, further experimental systems to manipulate the nature of cell cycle transitions as well as mathematical modeling are required to verify that DNA replication control critically requires sharp cell cycle transitions.

### 3.2. Temporal Order of Licensing/Firing Activation/Inactivation at Cell Cycle Transitions

With cell cycle regulators undergoing switch-like transitions, one can speculate that the activation and inactivation of licensing and firing factors would also be switch-like (Figure 2A). With switch-like licensing–firing transitions, it would, for example, be possible to simultaneously activate firing and inactivate licensing at G1/S without risking over-replication. At this point, we can, however, not exclude the possibility that the regulation of replication control factors will be more gradual, depending, for example, on slow turnover of proteins or their modifications (Figure 2B). With gradual licensing–firing transitions, a simultaneous activation/inactivation could potentially involve licensing and firing factors being (partially) active at the same time (Figure 2B), therefore potentially allowing sporadic over-replication. 

Currently, there is little evidence supporting either of the two models. There is, however, mounting evidence that activation/inactivation of licensing/firing factors does not occur simultaneously, but that it is ordered (Figure 2C). This order appears to follow the general “inactivate first, activate second” rule, which means that at the G1/S transition licensing is turned off before firing is activated and that at the metaphase-to-anaphase transition firing is turned off before licensing is activated. This control will therefore result in temporal gaps between licensing and firing phases, which could be a mechanism to provide a robust block to over-replication. In the following section, we summarize our knowledge of mechanisms (Figure 3) potentially generating such temporal order. It needs to be pointed out, however, that at this stage, we have only isolated pieces of evidence for individual mechanisms, primarily from studies in budding yeast and the *Xenopus* system. Therefore, while a general picture is emerging, we cannot yet comment on the conservation of individual mechanisms.

### 3.3. Intrinsic Temporal Order by CDK–Substrate Interactions

Differential affinities of CDK to its substrates involved in licensing and firing could lead to those substrates becoming phosphorylated at different kinase concentrations. With rising CDK activity, such a mechanism could lead to a temporal offset in the inactivation of licensing factors and the activation of firing factors. Notably, both licensing and firing factors are also subject to multisite phosphorylation [5,48,49,77,78,85]. It is therefore easily conceivable that different thresholds are generated dependent on (i) the number of sites needed for activation/inactivation, (ii) the affinity of CDK for the individual sites, and (iii) the overall affinity for the substrate. Indeed, a mathematical model of replication initiation in budding yeast suggests that multisite phosphorylation of Sld2 (together with differential G1-CDK and S-CDK phosphorylation, see below) could cause licensing to become inactivated before firing becomes activated, thereby generating a temporal gap between the two processes [158]. 

An additional feature of CDK control that is utilized to generate order in the replication program is the fact that CDK is not a single entity, but that different cyclin–CDK complexes are active (to quantitatively different degrees) at different points throughout the cell cycle. Indeed, some budding yeast licensing factors are already targeted and inactivated by G1/S-CDK [101,102,104], while firing factors are only targeted and activated by S-CDK and not by G1/S-CDK [48,49,51,145]. G1/S-CDK activation precedes and is required for S-CDK activation, and mathematical modeling therefore suggests that the differential substrate specificity of G1/S-CDK and S-CDK for licensing and firing factors (together with effects by multisite phosphorylation, see above) will induce a delay in firing activation with respect to licensing inactivation [158]. Overall, these data suggest a temporal order at the G1/S transition in budding yeast, whereby licensing factors are inactivated first and firing factors are activated second. 

Differential specificity of different CDK complexes could also play a role at the metaphase-to-anaphase transition. While in budding yeast most licensing factors are better substrates for S-CDK (Clb5–Cdk1) than for M-CDK (Clb2–Cdk1), Mcm3 seems not to follow this general trend and to be very efficiently phosphorylated also by M-CDK [159]. Firing could therefore become inactivated as soon as levels of S-CDK drop in M phase, while licensing inhibition (via Mcm3 phosphorylation and Orc6 phosphorylation (see next paragraph)) is maintained as long as M-CDK is active, consistent with temporal order and the “inactivate first, activate second” model.

### 3.4. Temporal Order by Phosphatase–Substrate Interactions

At the metaphase-to-anaphase transition, replication initiation factors become dephosphorylated, which for firing factors corresponds to inactivation and for licensing factors corresponds to activation. While dephosphorylation could in principle also be achieved by protein turnover (see below), most replication factors appear to be actively dephosphorylated by protein phosphatases. It is therefore conceivable that phosphatases play a role in determining a temporal order of firing inactivation and licensing activation. One mitotic phosphatase that was shown to act on replication factors (Orc2, Orc6, Mcm3, Sld2, and Sld3) is budding yeast Cdc14 [145,160,161], which is a specific antagonist of CDK phosphorylation function during mitotic exit [162]. Notably, it was shown that Cdc14 has differential catalytic efficiency for different substrates, which results in temporally ordered dephosphorylation [163]. Interestingly, Orc6 dephosphorylation occurred late compared to other Cdc14 substrates [163], apparently generating a window of opportunity for prior dephosphorylation of firing factors. It needs to be noted that even after Cdc14 inactivation, the temporal order of dephosphorylation of firing factors (first) and licensing factors (second) at the metaphase-to-anaphase transition remained largely intact [145]. While Cdc14 may therefore be a factor contributing to a temporal order of dephosphorylation at the metaphase-to-anaphase transition, it is not the only mechanism. Other phosphatases acting redundantly with Cdc14 or degradative mechanisms (see below) are likely candidates.

### 3.5. Temporal Order by Degradation

Cell cycle-regulated degradation is a commonly used mechanism to inactivate licensing factors (e.g., budding yeast Cdc6 [100,101,102,103] and Cdt1 in metazoa and fission yeast [120,121,122,123,124,125,126,164], see above) during the firing phase or to inactivate licensing inhibitors (e.g., metazoan geminin [117], see above) during the licensing phase. Additionally, cell cycle phase-specific degradation can be used to generate temporal order at cell cycle transitions.

A mechanism of early inactivation by degradation may be active at the G1/S transition in budding yeast. The licensing factor Cdc6, which is strongly regulated by degradation, including most importantly efficient inactivation by SCF–Cdc4 in S phase [100,101,102], is even a target for degradation in G1. Notably, with the exception of a possible involvement of the ubiquitin ligases Tom1 and Dia2, we currently have little knowledge of the mechanism and function of Cdc6 degradation in G1 (so called “mode 1” [101,103]). It is conceivable that enhanced turnover of Cdc6 prior to the G1/S transition may help Cdc6 inactivation.

Interestingly, a similar mechanism (early degradation) is active on the firing factor Sld2 before the metaphase-to-anaphase transition in budding yeast [145] and was also shown to be involved in meiotic DNA replication control [165]. Sld2 degradation at the metaphase-to-anaphase transition depends on CDK phosphorylation of Sld2 but additionally requires cell cycle kinases DDK, Cdc5, and Mck1, which phosphorylate Sld2 at a phospho-degron motif, which is read out by the ubiquitin ligases Dma1 and Dma2 [145]. Dependency on these four kinases generates a specific temporal window of Sld2 degradation in early M phase before the metaphase-to-anaphase transition. Enhanced Sld2 turnover at the metaphase-to-anaphase transition is apparently used to generate a temporal order of “inactivate firing first and activate licensing second”, as mutation of the Sld2 degron interferes with this order and shortens the gap between firing inactivation and licensing activation [145]. Notably, this mechanism is likely involved in preventing sporadic over-replication, as strains deficient in Sld2 degradation show an origin-dependent chromosomal rearrangement phenotype when combined with mutants that constitutively activate Sld3 [145]. Overall, both mechanisms leading to early degradation of Cdc6 and Sld2 can be envisioned to establish ordered inactivation of licensing/firing factors before activation of firing/licensing factors.

Notably, mammalian Cdc6 was also found to become degraded in G1-arrested cells by APC^Cdh1^ [166]. It seems, however, questionable whether this mechanism is involved in promoting early inactivation of Cdc6 prior to the G1/S transition, as Cdc6 becomes phosphorylated right at the G1/S transition by G1/S-CDK (cyclin E-Cdk2) [167]. This phosphorylation protects Cdc6 from APC-dependent degradation and induces a wave of licensing in late G1 [167]. While G1/S-CDK phosphorylation activates Cdc6, S-CDK (cyclin A-Cdk2) phosphorylation inhibits Cdc6 by promoting nuclear export [168,169,170]. Moreover, Cdc6 becomes degraded in S phase in a phosphorylation-independent manner [171]. With activating and inactivating phosphorylation marks and different modes of degradation, it is therefore at this point unclear how Cdc6 inactivation relates to the activation of firing factors.

Lastly, it is interesting to note that even substrates of the same ubiquitin ligase can display temporal order in the degradation. As such, it was shown for the APC that its substrates become degraded with temporal order ([172,173], see below). Similarly, the human CRL4–Cdt2 ubiquitin ligase, which specifically degrades “G1 factors” such as Cdt1 in early S phase in a manner that depends on the replication factor PCNA [120,122,123,164], targets different substrates to become degraded with different kinetics [174]. Interestingly, the licensing factor Cdt1 was shown to be degraded prior to p21, the critical S-CDK and G1/S-CDK inhibitor [174]. Importantly, according to our current understanding, degradation of both factors requires DNA-bound PCNA and therefore occurs after the first origins have already fired [122,123,164]. Nonetheless, these data suggest that licensing is inhibited due to Cdt1 degradation before S-CDK is fully active due to degradation of p21 [174]. This suggests that temporal ordering of degradation of different substrates by a single ubiquitin ligase may contribute to the overall temporal order of DNA replication control. 

### 3.6. Temporal Order by a Two-Kinase System

An apparently universal feature of eukaryotic DNA replication is the dual control by CDK and DDK. The underlying reason for this two-kinase system, however, remains obscure. While essential roles for two independent kinases acting in the temporal program of replication or the response to replication perturbation seem likely [96,175,176,177], we would like to speculate that such a two-kinase system may also be suited to generate temporal order at cell cycle transitions.

Specifically, it needs to be noted that licensing inhibition has so far exclusively been attributed to CDK [5,99,100,101,102,104], while firing activation essentially requires both CDK and DDK [48,49,51,74,81,82,83,85]. Therefore, if CDK was activated before DDK at the G1/S transition, licensing would be inhibited before firing was activated, thus minimizing the risk of over-replication. Indeed, DDK activity is cell cycle regulated and kept low in G1 by APC-mediated degradation of Dbf4 [63,66,178]. However, when compared to CDK, overall changes in global DDK activity are not as pronounced [172]. Rather, the DDK activity profile appears to be sharpened locally by DDK inhibition at replication origins via the antagonizing Rif1–PP1 phosphatase complex [177,179,180] and via antagonizing Mcm2–7 modification with SUMO [181]. Currently, it is therefore unclear whether DDK is indeed activated after S-CDK. 

However, genetic arguments from studies in budding yeast support that DDK might limit the timing of firing: (i) While previous studies have indicated that in the order of molecular events during firing the DDK-dependent step precedes the CDK-dependent steps [31,37,97], it was derived from conditional mutations in CDK or DDK that DDK action requires prior CDK activity [96,182]. (ii) DDK was found to be a limiting factor for origin firing [175] and it seems reasonable that the rather low amounts of DDK kinase will phosphorylate replication factors intrinsically more slowly than abundant CDK. In contrast, data from in vitro reconstitution using extracts or purified proteins suggests that DDK- and CDK-dependent steps are not necessarily separate [31,37]. Indeed, firing can be experimentally induced no matter whether replication proteins are first phosphorylated with CDK or DDK, respectively [37]. Nonetheless, these data would still be consistent with a temporal order of rising DDK and CDK activity at the G1/S transition in vivo.

Lastly, is the converse true in M phase? Is DDK inactivated prior to CDK at the metaphase-to-anaphase transition? Indeed, a recent study in budding yeast suggests that the APC mediates degradation of its key substrates (the anaphase inhibitor securin, S-phase cyclin Clb5, M-phase cyclin Clb2 and Dbf4) with a specific order and, indeed, Dbf4 was found to become degraded prior to Clb2, suggesting that at the metaphase-to-anaphase transition, DDK activity will drop prior to CDK activity [172]. This finding is therefore consistent with firing inactivation occurring prior to licensing reactivation in M phase and suggests that an overall purpose of the two-kinase control of replication might be temporal separation of licensing and firing. 

## 4. Conclusions and Outlook

The principal control ensuring temporal separation of the two phases of replication initiation and thereby “once-per-cell-cycle replication” are well understood in several eukaryotic models. On top of this control exist several auxiliary mechanisms which appear to act specifically at cell cycle transitions between the two phases of replication initiation. Their sheer existence suggests that additional control is needed to equip DNA replication with the formidable accuracy that eukaryotes require to propagate a stable genome over generations. We propose that a specific purpose of these mechanisms is to generate temporal order in the inactivation and activation of licensing and firing factors at cell cycle transitions in order to provide a robust block against sporadic over-replication events which, otherwise, may manifest in genome instability [6,8,127,128,130,145,147]. Detailed studies of these mechanisms are needed to fill the gaps in our knowledge on how cells provide smooth transitions from licensing to firing phases and vice versa. In particular, we need to address scenarios of deregulated replication initiation leading to sporadic over-replication in order to reveal whether and to what degree single cells and, in particular, organisms can tolerate sporadic over-replication and whether sporadic over-replication could be a driving force in cancer. 

## Figures and Tables

**Figure 1 genes-10-00099-f001:**
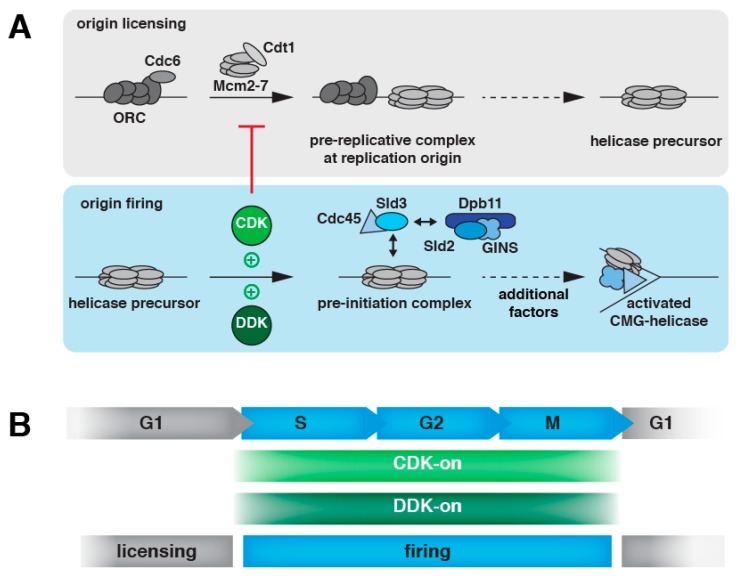
Two-step mechanism of DNA replication initiation. (**A**) Inactive helicase precursors are loaded during origin licensing (upper panel); CDK and DDK promote activation of these precursors to form active CMG helicases during origin firing (lower panel). In addition to the depicted factors, origin firing and helicase activation involve Sld7, DNA polymerase ε, and Mcm10, which are indicated as additional factors. (**B**) Changing activity of CDK and DDK couples licensing and firing strictly to distinct phases of the cell cycle.

**Figure 2 genes-10-00099-f002:**
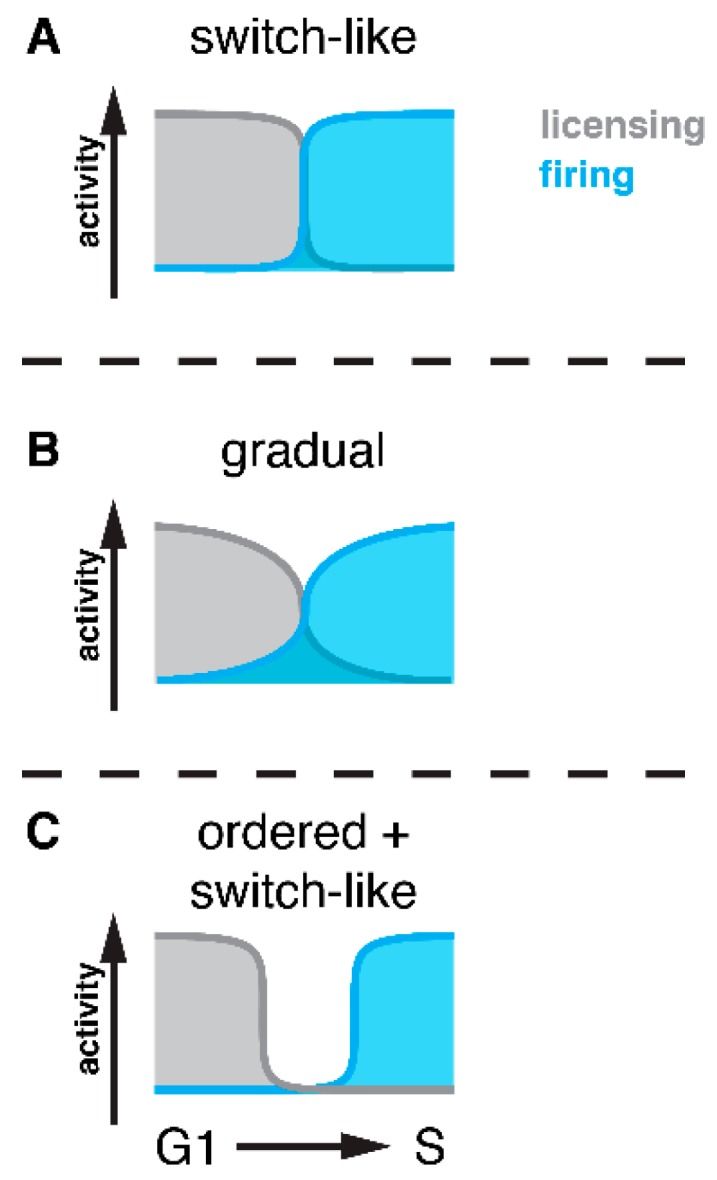
Separation of licensing and firing at cell cycle phase transitions. Three different models for transitions between replication phases are depicted here for the example of the G1/S transition. Note the temporal overlap of licensing (grey) and firing (blue) activity in the different models, indicating the potential for sporadic over-replication.

**Figure 3 genes-10-00099-f003:**
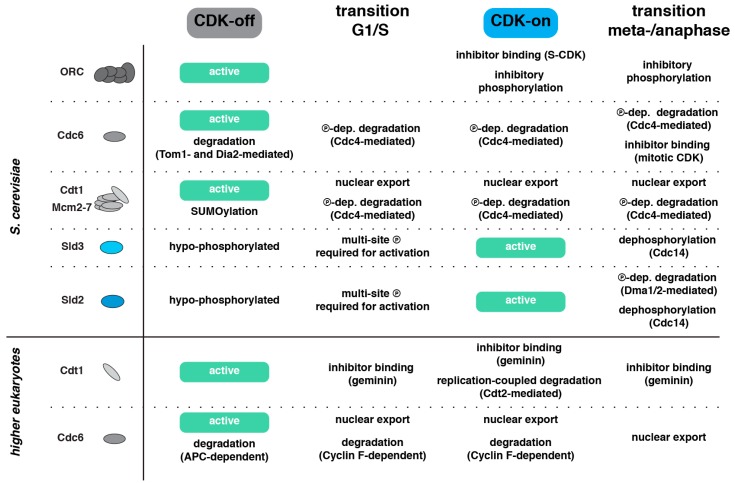
Mechanisms preventing over-replication. Summary of molecular mechanisms inhibiting licensing or firing at specific cell cycle phases (CDK-off or CDK-on) and specifically at the G1/S transition and the metaphase-to-anaphase transition. These inhibitory mechanisms are likely candidates to generate a temporal order in the activation/inactivation of licensing and firing factors and to thereby counteract sporadic over-replication at cell cycle transitions and limit replication to once per cell cycle.

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
