# Peer review of "Control of Eukaryotic DNA Replication Initiation—Mechanisms to Ensure Smooth Transitions"

_genes, 2019, doi:10.3390/genes10020099_

Round 1

Reviewer 1 Report

Reusswig and Pfander reviewed recent progress in regulation of eukaryotic replication initiation. Manuscript is generally well written and will be recommended for publication after the following concerns are considered.

Lines 160 and onwards
This part describe regulatory mechanisms mainly in budding yeast and Xenopus. Authors may pay more attention to what are conserved and what are different among eukaryotes.

Figure 1B
CDK and DDK acitivies go up and down at the same time at similar speed. Is this correct? Authors discuss order of CDK and DDK later (page 12) and therefore this figure is somewhat confusing.

Figure 1A (and 3)
Cartoons do not represent recent biochemical and structural studies. Suggested revisions include, but are not limited to, assignments of subunits where available, positions of Orc1 and Orc6 in ORC, DNA bending at central channel of ORC, correction of size (e.g., too big Cdc6), positions of OCCM subunits and pre-IC proteins, addition of Sld7 and pol epsilon, and indication of 5' and 3'. Also, there are no papers describing ORC-Cdc6-Cdt1-MCM double hexamer.

Minor comments

Text
Italicize where necessary.

Title
Title looks too general and misleading as if multiple domains of life were described.

Lines 22 and 23
Cite literature.

Author Response

We thank the reviewers for commending our work for publication, as well as for their constructive and very helpful criticism. As seen in our response below, we have done our best to incorporate their points as well as possible in our revised version of the manuscript. Please find our point-by-point response below (response in blue and italicised).

reviewer #1

Reusswig and Pfander reviewed recent progress in regulation of eukaryotic replication initiation. Manuscript is generally well written and will be recommended for publication after the following concerns are considered.

We thank the reviewer for his/her encouraging words. 

Lines 160 and onwards
This part describe regulatory mechanisms mainly in budding yeast and Xenopus. Authors may pay more attention to what are conserved and what are different among eukaryotes.

We clarified that most of our knowledge stems from isolated studies in model organisms and that – particularly in case of mechanism ensuring temporal order at transitions – we can so far not conclude whether specific mechanisms are conserved among different eukaryotes.

Figure 1B
CDK and DDK acitivies go up and down at the same time at similar speed. Is this correct? Authors discuss order of CDK and DDK later (page 12) and therefore this figure is somewhat confusing.

Thanks for pointing this out. We now change the graphical depiction in Fig. 1B so that it does not make a statement on the relative activation/inactivation kinetics of CDK and DDK.

Figure 1A (and 3)
Cartoons do not represent recent biochemical and structural studies. Suggested revisions include, but are not limited to, assignments of subunits where available, positions of Orc1 and Orc6 in ORC, DNA bending at central channel of ORC, correction of size (e.g., too big Cdc6), positions of OCCM subunits and pre-IC proteins, addition of Sld7 and pol epsilon, and indication of 5' and 3'. Also, there are no papers describing ORC-Cdc6-Cdt1-MCM double hexamer.

We agree that our depiction of the licensing reaction could be misleading and have therefore changed Fig. 1A to avoid any confusion. We have introduce “other factors”, which we define in the legend to account for proteins such as polymerase epsilon and Sld7. Other than that we do not aim to represent molecular detail.

Minor comments

Text
Italicize where necessary.

Done.

Title
Title looks too general and misleading as if multiple domains of life were described.

We added “eukaryotic” to the title, in order not to be misleading. Thank you.

Lines 22 and 23
Cite literature.

We introduced the citation „DNA replication control occurs with exceptional accuracy to keep the genetic information stable over as many as 1016 cell divisions (estimations based on [1])  during, for example, the average human lifespan.“

Reviewer 2 Report

Comments on genes-428553 (Karl-Uwe Reusswig and Boris Pfander, Control of DNA replication initiation – mechanisms to ensure smooth transitions).

In this review article, authors review how eukaryotic cells ensure genome integrity by preventing over-replication, especially by focusing the temporal separation of licensing and firing reaction in the initiation of DNA replication. Generally, this review is well-written and therefore worth to publish in the genes. However, I feel there are a couple of issues to be clarified before publication. If they are solved, I recommend the publication of this review article. Specific points are follows.

Major points

1. Authors describe licensing and firing are ‘interdependent’ (lines 36, 50, and 86). I do not think they are interdependent. Of course, firing depends on preceding licensing, but the converse is not true.

2. Authors have forgot to include DNA polymerase epsilon (Pol ε) in firing factors (please see: Sengupta et al (2013) Curr Biol 23, 543- (Dpb2, in vivo); Miyazawa-Onam et al (2017) EMBO Rep 18, 1752- (Pol2, in vivo); and Yeeles et al (2015) Nature 519, 431- (in vitro)) (line 61, and 115). Instead, they included RPA, I do not think RPA is required for the firing reaction, which indicates helicase activation. Figure 1A also should include Pol ε.

3. When they describe the experimental over-replication system which includes the failure of the separation of licensing and firing, they only describe the de-regulation of licensing control throughout the review. However, it has already shown that the de-regulation of firing also causes over-replication, and the following DNA damage occurrence and genome instability (Tanaka et al (2017) Nature 445: 328-, Zegerman and Diffley (2007) Nature 445: 281-, and Tanaka and Araki (2011) PLoS Genet. 7, e1002136.). Authors should refer those works for readers. 

4. The difference between the section ‘Deregulation of DNA replication initiation – over-replication and genome instability’ (lines 160-195) and the following section ‘Partial deregulation of DNA replication initiation – sporadic over-replication’ (lines 196-223) was unclear for me. What is the mechanical difference? JJLi’s group has done many experiments combining de-regulation of licensing factors (Green et al (2006) Mol Biol Cell 17, 2401-). In this work, only limited region of one chromosome was over-replicated, when Mcm2-7 and Cdc6 are deregulated. Is this not a ‘partial deregulation’ and ‘sporadic over-replication’? If authors more precisely define the ‘partial deregulation’ and the ‘sporadic over-replication’, it would be very nice. 

Authors also describe DNA replication is regulated in ‘multi-layered’ way (line 206). Please clarify this ‘multi-layer’ by mechanistic level to remove the ambiguity of the description.

Minor points

1. When authors describe explain temporal gap between licensing and firing in the later sections, they should first describe Cln activation turns off the licensing before Clb activation, which turns on the firing. Mechanistically, Clb activation fully depends on Cln activation, which ensures switch-off of licensing precedes switch-on of firing and is important design principle in the cell cycle. If authors describe this principle first, it may greatly help the understanding of readers.

2. In Figure 1A top, authors include Cdc6 and Cdt1 in pre-replicative complex, pre-RC. I do not think this idea is not popular now, and I think many people think ORC and Mcm2-7 double hexamer, or Mcm2-7 double hexamer alone are the entity of pre-RC. 

3. In the last section: ‘Temporal order by a two-kinase system’, authors should clearly describe DDK-dependent process precedes CDK-dependent process in the firing mechanism, as many works have clearly shown. As authors describe, both DDK and CDK are active in S phase cells and the phosphorylation of kinase substrates is sufficient for the firing, however, authors should clearly describe the firing mechanism for readers. 

3. Lines 261-262: ‘which is consistent with the occurrence of sporadic over-replication at the G1/S transition’. Please show reference(s).

4. Nguyen et al (2000) Curr. Biol. 10: 195–205. should be added as references (line 125).

5.Tanaka et al (2011)Curr. Biol. 21: 2055-2063. should be added as references (lines 396 and 411). Because Mantiero et al (2011, EMBO J. 30: 4805-) and Tanaka et al shows the same results, although the combination of replication factors are slightly different.

6. Closing parenthesis is missing (line 57).

Author Response

We thank the reviewers for commending our work for publication, as well as for their constructive and very helpful criticism. As seen in our response below, we have done our best to incorporate their points as well as possible in our revised version of the manuscript. Please find our point-by-point response below (response in blue and italicised).

reviewer #2

In this review article, authors review how eukaryotic cells ensure genome integrity by preventing over-replication, especially by focusing the temporal separation of licensing and firing reaction in the initiation of DNA replication. Generally, this review is well-written and therefore worth to publish in the genes. However, I feel there are a couple of issues to be clarified before publication. If they are solved, I recommend the publication of this review article. Specific points are follows.

We thank the reviewer for his/her attention to detail and for the constructive criticism. 

Major points

1. Authors describe licensing and firing are ‘interdependent’ (lines 36, 50, and 86). I do not think they are interdependent. Of course, firing depends on preceding licensing, but the converse is not true.

Agreed. Licensing does not require preceding firing. We have changed “interdependent” to “interconnected”.

2. Authors have forgot to include DNA polymerase epsilon (Pol ε) in firing factors (please see: Sengupta et al (2013) Curr Biol 23, 543- (Dpb2, in vivo); Miyazawa-Onam et al (2017) EMBO Rep 18, 1752- (Pol2, in vivo); and Yeeles et al (2015) Nature 519, 431- (in vitro)) (line 61, and 115). Instead, they included RPA, I do not think RPA is required for the firing reaction, which indicates helicase activation. Figure 1A also should include Pol ε.

Agreed. We did not want to omit the initiation role of Pol epsilon and have included it in the text as well as in Fig. 1A (as additional factors, which is explained in the legend). We have also included the necessary citations.

3. When they describe the experimental over-replication system which includes the failure of the separation of licensing and firing, they only describe the de-regulation of licensing control throughout the review. However, it has already shown that the de-regulation of firing also causes over-replication, and the following DNA damage occurrence and genome instability (Tanaka et al (2017) Nature 445: 328-, Zegerman and Diffley (2007) Nature 445: 281-, and Tanaka and Araki (2011) PLoS Genet. 7, e1002136.). Authors should refer those works for readers. 

Indeed, we think that systems to de-regulate firing, such as those developed by the Diffley and (in particular) Araki labs are very interesting. We have therefore also mentioned these systems (in particular the 2011 PLoS Genetics paper), when it comes to systems that appear to induce sporadic over-replication. And all papers are cited multiple times in our review. We have however not put a specific focus on systems to de-regulate firing, as this would take away from the specific focus of our review, namely mechanisms that specifically act at cell cycle transitions.

4. The difference between the section ‘Deregulation of DNA replication initiation – over-replication and genome instability’ (lines 160-195) and the following section ‘Partial deregulation of DNA replication initiation – sporadic over-replication’ (lines 196-223) was unclear for me. What is the mechanical difference? JJLi’s group has done many experiments combining de-regulation of licensing factors (Green et al (2006) Mol Biol Cell 17, 2401-). In this work, only limited region of one chromosome was over-replicated, when Mcm2-7 and Cdc6 are deregulated. Is this not a ‘partial deregulation’ and ‘sporadic over-replication’? If authors more precisely define the ‘partial deregulation’ and the ‘sporadic over-replication’, it would be very nice. 

Authors also describe DNA replication is regulated in ‘multi-layered’ way (line 206). Please clarify this ‘multi-layer’ by mechanistic level to remove the ambiguity of the description.

We entirely agree that the partial de-regulation of licensing factors by the Li lab, as well as partial de-regulation of firing factors by the Araki lab and us can be interpreted as giving different degrees of sporadic over-replication and we now mention and cite those works in the corresponding paragraphs. The principal problem that we wanted to raise in this part is the experimental difficulty in studying these systems. We have very sensitive assays to measure genome rearrangements, but other assays are not as sensitive. Sporadic over-replication systems from us and the Araki lab basically gave sporadic over-replication as the major phenotype. The Li over-replication system has the advantage of affecting one origin in particular, which therefore could be better studied, but it therefore probably also was less “sporadic”.

We have also clarified the usage of “multi-layered”. 

Minor points

1. When authors describe explain temporal gap between licensing and firing in the later sections, they should first describe Cln activation turns off the licensing before Clb activation, which turns on the firing. Mechanistically, Clb activation fully depends on Cln activation, which ensures switch-off of licensing precedes switch-on of firing and is important design principle in the cell cycle. If authors describe this principle first, it may greatly help the understanding of readers.

Agreed. We have now included a sentence that states that activation of G1-CDKs not only precedes activation of S-CDKs, but importantly that activation of S-CDKs requires activation of G1-CDKs.

2. In Figure 1A top, authors include Cdc6 and Cdt1 in pre-replicative complex, pre-RC. I do not think this idea is not popular now, and I think many people think ORC and Mcm2-7 double hexamer, or Mcm2-7 double hexamer alone are the entity of pre-RC. 

Agreed. We understand that our initial depiction of the pre-RC was misleading and have changed this in the new version of the review.

3. In the last section: ‘Temporal order by a two-kinase system’, authors should clearly describe DDK-dependent process precedes CDK-dependent process in the firing mechanism, as many works have clearly shown. As authors describe, both DDK and CDK are active in S phase cells and the phosphorylation of kinase substrates is sufficient for the firing, however, authors should clearly describe the firing mechanism for readers.

Agreed. We have now highlighted the interesting contradiction that on the one hand the CDK-dependent steps during firing appear to come after the DDK-dependent steps, while on the other hand DDK appears to become activated only after S-CDK. We think that this strongly suggests that DDK activation poses a bottleneck to origin firing.  

3. Lines 261-262: ‘which is consistent with the occurrence of sporadic over-replication at the G1/S transition’. Please show reference(s).

The second half-sentence is our interpretation based on our previous work and previous work from the Araki and Li labs. We think a citation would not be suitable here and have changed the sentence to “…which can be interpreted to be a consequence of sporadic over-replication at the G1/S transition.”  

4. Nguyen et al (2000) Curr. Biol. 10: 195–205. should be added as references (line 125).

Added the reference.

5.Tanaka et al (2011)Curr. Biol. 21: 2055-2063. should be added as references (lines 396 and 411). Because Mantiero et al (2011, EMBO J. 30: 4805-) and Tanaka et al shows the same results, although the combination of replication factors are slightly different.

Added the reference.

6. Closing parenthesis is missing (line 57).

Corrected.

Reviewer 3 Report

This review by Reusswig and Pfander provides a comprehensive review of how eukaryotic cells maintain once per cell cycle replication of their DNA. They cover all the established concepts of how the processes of origin licencing and origin firing are kept temporally separate. They also provide a nice section exploring how the affinities of CDK substrates involved in licencing and firing for phosphorylation and dephosphorylation through phosphatase differ in a manner that further ensures that the processes of licencing and firing do not overlap. Other than a couple of typographic errors I have no issues with the information presented and can happily recommend it for publication in Genes.

Minor errors

1)    In the abstract the authors state “DNA replication initiation therefore underlies exquisite control”. My sense is that they mean to say that “DNA replication initiation is exquisitely controlled”

2)    In line 343 they say ”… Cdt1 in metazoa and fissing yeast”. I assume they mean “ in metazoa fission yeast”

Author Response

We thank the reviewers for commending our work for publication, as well as for their constructive and very helpful criticism. As seen in our response below, we have done our best to incorporate their points as well as possible in our revised version of the manuscript. Please find our point-by-point response below (response in blue and italicised).

reviewer #3

This review by Reusswig and Pfander provides a comprehensive review of how eukaryotic cells maintain once per cell cycle replication of their DNA. They cover all the established concepts of how the processes of origin licencing and origin firing are kept temporally separate. They also provide a nice section exploring how the affinities of CDK substrates involved in licencing and firing for phosphorylation and dephosphorylation through phosphatase differ in a manner that further ensures that the processes of licencing and firing do not overlap. Other than a couple of typographic errors I have no issues with the information presented and can happily recommend it for publication in Genes.

We thank the reviewer for his/her encouraging words. 

Minor errors

1)    In the abstract the authors state “DNA replication initiation therefore underlies exquisite control”. My sense is that they mean to say that “DNA replication initiation is exquisitely controlled”

done

2)    In line 343 they say ”… Cdt1 in metazoa and fissing yeast”. I assume they mean “ in metazoa fission yeast”

corrected